# Insight into the Natural History of Pathogenic Variant c.919-2A>G in the *SLC26A4* Gene Involved in Hearing Loss: The Evidence for Its Common Origin in Southern Siberia (Russia)

**DOI:** 10.3390/genes14040928

**Published:** 2023-04-17

**Authors:** Valeriia Yu. Danilchenko, Marina V. Zytsar, Ekaterina A. Maslova, Konstantin E. Orishchenko, Olga L. Posukh

**Affiliations:** 1Federal Research Center Institute of Cytology and Genetics, Siberian Branch of the Russian Academy of Sciences, 630090 Novosibirsk, Russia; danilchenko_valeri@mail.ru (V.Y.D.); zytzar@bionet.nsc.ru (M.V.Z.); maslova@bionet.nsc.ru (E.A.M.); keor@bionet.nsc.ru (K.E.O.); 2Novosibirsk State University, 630090 Novosibirsk, Russia

**Keywords:** hearing loss, *SLC26A4*, c.919-2A>G, founder effect, STR and SNP haplotypes, Tuvinians

## Abstract

Pathogenic variants in the *SLC26A4* gene leading to nonsyndromic recessive deafness (DFNB4), or Pendred syndrome, are some of the most common causes of hearing loss worldwide. Earlier, we found a high proportion of *SLC26A4*-related hearing loss with prevailing pathogenic variant c.919-2A>G (69.3% among all mutated *SLC26A4* alleles that have been identified) in Tuvinian patients belonging to the indigenous Turkic-speaking Siberian people living in the Tyva Republic (Southern Siberia, Russia), which implies a founder effect in the accumulation of c.919-2A>G in Tuvinians. To evaluate a possible common origin of c.919-2A>G, we genotyped polymorphic STR and SNP markers, intragenic and flanking *SLC26A4*, in patients homozygous for c.919-2A>G and in healthy controls. The common STR and SNP haplotypes carrying c.919-2A>G were revealed, which convincingly indicates the origin of c.919-2A>G from a single ancestor, supporting a crucial role of the founder effect in the c.919-2A>G prevalence in Tuvinians. Comparison analysis with previously published data revealed the identity of the small SNP haplotype (~4.5 kb) in Tuvinian and Han Chinese carriers of c.919-2A>G, which suggests their common origin from founder chromosomes. We assume that c.919-2A>G could have originated in the geographically close territories of China or Tuva and subsequently spread to other regions of Asia. In addition, the time intervals of the c.919-2A>G occurrence in Tuvinians were roughly estimated.

## 1. Introduction

Pathogenic variants in the *SLC26A4* gene (solute carrier family 26, member 4/pendrin, 7q22.3, OMIM 605646) are one of the most common causes of hearing loss worldwide. The *SLC26A4* gene (21 exons) encodes the protein pendrin, which is involved in the transport of various anions [1]. High levels of *SLC26A4* expression are observed in the inner ear, thyroid, and kidneys [2]. Several hundred pathogenic *SLC26A4* variants (the Deafness Variation Database: https://deafnessvariationdatabase.org/gene/SLC26A4, accessed on 13 February 2023) are currently known to be associated with varying phenotypes. They can lead to nonsyndromic recessive deafness (DFNB4) or Pendred syndrome. The DFNB4 (OMIM 600791) is characterized by the prelingual or perilingual onset of sensorineural or mixed hearing loss, which may be fluctuating or progressive. Pendred syndrome (PDS, OMIM 274600) is an autosomal recessive disorder associated with sensorineural hearing loss and goiter. In the inner ear, deficiency or dysfunction of pendrin presumably leads to the development of endolymphatic hydrops due to defects in anion and fluid transport. As a result, *SLC26A4*-related hearing loss is most commonly accompanied by the enlarged vestibular aqueduct (EVA) and/or other malformations of the inner ear structures [3].

At present, numerous studies aimed to investigate the prevalence of *SLC26A4*-related hearing loss, as well as the distribution of pathogenic *SLC26A4* variants in various regions of the world. The spectrum of *SLC26A4* pathogenic variants found in Asian populations appears to differ from that in populations of Caucasian origin: the variants c.919-2A>G and c.2168A>G (p.His723Arg) are the most common in East Asia, while they are very rare or absent in Europe; variants c.1001+1G>A, c.412G>T (p.Val138Phe), c.1246A>C (p.Thr416Pro), c.707T>C (p.Leu236Pro), and c.626G>T (p.Gly209Val) are prevalent in many Caucasian populations [4,5,6,7,8,9,10]. Only several *SLC26A4* pathogenic variants (c.1226G>A (p.Arg409His), c.1229C>T (p.Thr410Met), c.1334T>G (p.Leu445Trp), and c.1790T>C (p.Leu597Ser)) are ubiquitously found (with varying frequencies) in all regions of the world. The accumulation of a number of specific pathogenic *SLC26A4* variants in certain populations was suggested to be a result of the founder effect, as evidenced by conservation of haplotypes formed by closely linked genetic markers: STRs (short tandem repeats) or SNPs (single nucleotide polymorphisms). Common specific haplotypes were found for several recurrent pathogenic *SLC26A4* variants: c.2168A>G (p.His723Arg) in Japanese and Koreans; c.919-2A>G in Chinese [8,11,12]; c.412G>T (p.Val138Phe) in German patients [13]; c.1541A>G (p.Gln514Arg) in Spanish patients [14]; c.965dup (p.Asn322LysfsTer8) in Iranian patients [15]; and c.269C>T (p.Ser90Leu), c.716T>A (p.Val239Asp), and c.1337A>G (p.Gln446Arg) in families from Pakistan [10,16].

The c.919-2A>G variant (rs111033313) was shown to be recurrent in multiple East Asian populations [7,8,11]. This variant (originally named 1143-2A>G, later IVS7-2A>G) was firstly found in a Turkish family with Pendred syndrome [17]. The c.919-2A>G is located at the canonical acceptor splice site -2 in the intron region between exons 7 and 8 and leads to a skipping of exon 8. The deletion of exon 8 generates a new stop codon at position 311, which results in a premature truncated protein of only 310 amino acids [18].

Subsequently, in numerous studies, c.919-2A>G has been frequently identified in patients from Asian countries (mainland China and Taiwan, Mongolia, Korea, and Japan) and observed with the highest frequency in China, while c.919-2A>G is very rare or absent in other countries [4,5,8,12,19,20,21,22,23,24,25,26]. The observed frequencies of c.919-2A>G in global populations, according to the Genome Aggregation Database (gnomAD, https://gnomad.broadinstitute.org/, accessed on 13 February 2023), are as follows: 0.005378 in East Asian; 0.00001470 in European (non-Finnish); and 0.0 in South Asian, European (Finnish), Ashkenazi Jewish, Middle Eastern, Amish, African/African American, Latino/Admixed American, and Other.

In our recent study [27], we performed a thorough analysis of the *SLC26A4* gene by Sanger sequencing in the large cohorts of patients with hearing loss belonging to two neighboring indigenous Turkic-speaking Siberian peoples (Tuvinians and Altaians) (in the Tyva Republic and the Altai Republic, Southern Siberia, Russia). We found that 28.2% (62/220) of enrolled Tuvinian patients from the Tyva Republic (Tuva) had biallelic pathogenic *SLC26A4* variants. This rate of the *SLC26A4*-related hearing loss in Tuvinian patients appeared to be one of the highest among populations worldwide. The majority of Tuvinian patients were homozygous or compound heterozygous for c.919-2A>G. The proportion of this variant was 69.3% (95/137) among all *SLC26A4* mutant alleles identified in Tuvinian patients, and its carrier frequency in the Tuvinian population was 5.1% (8/157) [27].

A high rate of c.919-2A>G in Tuvinians implies a crucial role of the founder effect in its prevalence in this indigenous Siberian people. In this regard, we aimed to test a presumable common origin of c.919-2A>G in Tuvinians by analyzing the genetic background (haplotypes) of c.919-2A>G in the carriers of this *SLC26A4* pathogenic variant.

## 2. Materials and Methods

### 2.1. Subjects

Genotyping of genetic markers (STRs and SNPs) for the haplotype analysis was carried out in the sample of unrelated Tuvinian patients with hearing loss who were homozygous for variant c.919-2A>G (*n* = 23) and in the ethnically matched control sample (Tuvinians), which was represented by unrelated healthy individuals without c.919-2A>G (*n* = 63). Both samples were formed after our recent analysis of the *SLC26A4* gene in Tuvinians belonging to indigenous population of the Tyva Republic (Southern Siberia, Russia) [27].

### 2.2. Ethics Statement

Written informed consent was obtained from all individuals or their legal guardians before they participated in the study. The study was conducted in accordance with the Declaration of Helsinki, and the protocol was approved by the Bioethics Commission at the Institute of Cytology and Genetics SB RAS, Novosibirsk, Russia (Protocol No. 9, 24 April 2012).

### 2.3. STRs and SNPs Genotyping

To analyze the c.919-2A>G genetic background, we genotyped five STRs in the region of chromosome 7, including four STRs flanking the *SLC26A4* gene at different physical distances: centromeric D7S2420 (~0.43 Mb) and D7S496 (~0.17 Mb); telomeric D7S2456 (~0.36 Mb) and D7S525 (~2.32 Mb); and intragenic D7S2459, located approximately 7.6 kb away from c.919-2A>G. These STRs have been previously used for linkage analysis to define the genetic interval linked to Pendred syndrome or DFNB4 and haplotype analysis in analyzed pedigrees [15,28,29,30,31,32,33]. The total length of the region flanked by distal markers D7S2420 and D7S525 was approximately 2.8 Mb.

To study the fine structure of haplotypes including c.919-2A>G, nine intragenic SNPs (rs2248464, rs2248465, rs3801943, rs2712212, rs2395911, rs2712211, rs3801940, rs2072064, and rs2072065) that closely flanked c.919-2A>G were also genotyped. The choice of analyzed SNPs was based on their physical distances to c.919-2A>G and the variability (a minor allele frequency greater than 0.1) in global populations according to the Genome Aggregation Database (gnomAD, https://gnomad.broadinstitute.org/, accessed on 13 February 2023) (Appendix A). Four of them, rs2712212, rs2395911, rs2712211, and rs3801940, were included for comparative analysis with the data from the study by Wu et al. [12], where these SNPs (designated JST160568, JST089508, JST160566, and JST160565, respectively) were used to detect evidence of the founder effect for the c.919-2A>G variant in Taiwanese hearing-impaired patients [12]. The locations and physical distances of analyzed SNPs to c.919-2A>G were as follows: centromeric rs2248464 (intron 2, 20.28 kb), rs2248465 (intron 2, 20.27 kb), rs3801943 (intron 6, 2.23 kb), and rs2712212 (intron 6, 2.18 kb) and telomeric rs2395911 (intron 8, 0.22 kb), rs2712211 (intron 8, 2.02 kb), rs3801940 (intron 8, 2.32 kb), rs2072064 (intron 10, 10.62 kb), and rs2072065 (intron 10, 10.76 kb). The total length of the region flanked by the distal markers rs2248464 and rs2072065 was 31.039 kb.

The location of all analyzed genetic markers (STRs and SNPs) on chromosome 7 is presented in Figure 1.

The STRs genotyping was performed by fragment analysis. To amplify fragments containing STRs, the primer sequences were taken from the Ensembl genome browser (http://www.ensembl.org, accessed on 15 September 2022). One (forward) from each primer pair was marked on the 5′ end by different fluorescent dyes (Applied Biosystems 5´ Labeled/Unlabeled Primer Pairs, Thermo Fisher Scientific, Waltham, MA, USA). The SNP genotyping was performed by Sanger sequencing. To amplify fragments containing SNPs, the primer pairs were designed using Primer Premier 5 tools (https://www.bioprocessonline.com/doc/primer-premier-5-design-program-0001). All used primers are summarized in Appendix A. Fragment analysis and Sanger sequencing were performed in the SB RAS Genomics Core Facility (Institute of Chemical Biology and Fundamental Medicine SB RAS, Novosibirsk, Russia).

### 2.4. Reconstruction of STR and SNP Haplotypes

The STR and SNP genotyping data were used for the reconstruction and calculation of haplotype frequencies performed by the Expectation–Maximization (EM) algorithm of the Arlequin 3.5.1.2 software [34]. Linkage disequilibrium between the STR and SNP alleles and the c.919-2A>G variant was calculated using δ = (Pd − Pn)/(1 − Pn), where δ is the measure of linkage disequilibrium; Pd is the marker allele frequency among mutant chromosomes carrying c.919-2A>G (the sample of patients homozygous for c.919-2A>G); and Pn is the frequency of the same allele among normal chromosomes (control sample) [35].

### 2.5. Estimation of c.919-2A>G Age

The estimation of the c.919-2A>G age was performed by the DMLE+ v2.3 software method (Disequilibrium Mapping and Likelihood Estimation, http://dmle.org/) [36] and by the single-marker method using the algorithm [37] g = log [1 − Q/(1 − Pn)]/log(1 − Ѳ), where g is the number of generations passed from the moment of the mutation appearance to the present; Q is the share of mutant chromosomes unlinked with the founder haplotype; Pn is the population frequency of allele included in the founder haplotype; and Ѳ is the recombinant fraction calculated from physical distance between marker and mutation (under the assumption that 1 cM = 1000 kb). (See details in Appendix A).

### 2.6. Statistical Analysis

Fisher’s exact test with a significance level of *p* < 0.05 was applied to compare the allele and haplotype frequencies between the examined samples of patients and controls.

## 3. Results

We assumed that the high prevalence of c.919-2A>G in the *SLC26A4* gene in Tuvinians is a consequence of the founder effect. To test whether c.919-2A>G shares a common haplotype, we performed genotyping of polymorphic genetic markers: five STRs (four of them are flanking the *SLC26A4* gene and one is intragenic) and nine intragenic SNPs closely linked to c.919-2A>G in 23 unrelated Tuvinian patients homozygous for c.919-2A>G. We also genotyped the same genetic markers in 63 healthy unrelated Tuvinians without c.919-2A>G. The results of the STR and SNP genotyping in the sample of the c.919-2A>G homozygotes and in the control sample are summarized in Appendix A.

### 3.1. STR and SNP Haplotypes

**STR haplotypes**. Data on genotyping of five STRs (D7S2420, D7S496, D7S2459, D7S2456, and D7S525) were used to reconstruct STR haplotypes by the Expectation–Maximization (EM) algorithm of the Arlequin 3.5.1.2 software [34] both in Tuvinian deaf patients homozygous for c.919-2A>G and in the ethnically matched control sample. The boundaries of the shared STR haplotypes were determined by observed linkage disequilibrium between certain alleles of distal markers (D7S2420 and D7S525) and c.919-2A>G (Appendix A). The total length of the region flanked by D7S2420 and D7S525 is ~2.8 Mb. Four different haplotypes formed by specific alleles of all five STRs were found to be associated with c.919-2A>G in homozygotes for c.919-2A>G, while none of these haplotypes were detected in the control sample (Table 1). Among these haplotypes, the *278-120-147-244-227* haplotype was the most common (91.3%) among mutant chromosomes bearing c.919-2A>G.

**SNP haplotypes.** The SNP haplotypes were reconstructed based on the results of the genotyping of nine intragenic SNPs closely flanking c.919-2A>G (rs2248464, rs2248465, rs3801943, rs2712212, rs2395911, rs2712211, rs3801940, rs2072064, and rs2072065) (Table 1). Certain alleles of all analyzed SNPs showed strong linkage to c.919-2A>G (Appendix A), thereby forming the only specific haplotype A-C-T-A-G-G-C-A-C for all c.919-2A>G carriers (100%), while the frequency of this haplotype reached only 2.8% in the control sample (Table 1).

Four of the SNPs (rs2712212, rs2395911, rs2712211, and rs3801940) were early analyzed in the c.919-2A>G carriers from Taiwan in the study by Wu et al. [12], where the core haplotype T-C-C-G composed of certain alleles of these SNPs (designated in their study as JST160568, JST089508, JST160566, and JST160565, respectively) was revealed in a majority of chromosomes of the c.919-2A>G homozygotes, favoring the origin of c.919-2A>G from a common ancestor. In our study, when considering a haplotype constituted by these SNPs, a single haplotype (corresponding to T-C-C-G in the study by Wu et al. [12]) was found in all homozygotes for c.919-2A>G (100%) (Table 1), while this haplotype was the second by frequency (25.9%) among eight different SNP haplotypes found in the control sample.

### 3.2. Estimation of c.919-2A>G Age

Common STR and SNP haplotypes found for pathogenic *SLC26A4* variant c.919-2A>G, which is predominant in Tuvinians, suggest that c.919-2A>G originated from a single ancestor. We tried to evaluate the approximate “age” of c.919-2A>G by estimation of the numbers of generations (g) and years (with the assumption that g = 25 years) passed from the ancestral mutation event by the single-marker method (when appropriate) and by the DMLE+ v.2.3 program (Appendix A). Allele *227* of the distal STR marker D7S525 (~2.32 Mb from c.919-2A>G), which was found in strong linkage disequilibrium with c.919-2A>G (Appendix A), was used when applying the single-marker method. We were not able to apply the single-marker method for SNP markers due to the lack of recombination in all SNPs analyzed (Appendix A). The results of the c.919-2A>G age evaluation are summarized in Table 2.

The c.919-2A>G age estimations gave three time intervals depending on different population growth rates (d = 0.05, 0.1, and 0.2) that we applied for calculations (Table 2), thus demonstrating the sensitivity of the methods used from demographic parameters (Appendix A). In addition, we also calculated (using the DMLE+ v.2.3 program) the age of c.919-2A>G based on the SNP internal haplotype A-G-G-C (rs2712212-rs2395911-rs2712211-rs3801940) (Table 1). The variations in the c.919-2A>G age in that case, being 106–182 generations (2650–4550 years), 112–192 (2800–4800 years), and 105–188 generations (2625–4700 years) with d = 0.05, 0.1, and 0.2, respectively, indirectly indicate a more ancient age of this haplotype (Table 2).

## 4. Discussion

Understanding the regional or ethnospecific landscape of different pathogenic *SLC26A4* variants is still far from clear due to the heterogeneity in size and phenotypic characteristics of the examined cohorts of patients and the variable sensitivity of the *SLC26A4* molecular diagnostics in different studies. In particular, the proportion of c.919-2A>G, a well-known pathogenic variant, among other mutant alleles of the *SLC26A4* gene found in different cohorts of patients with *SLC26A4*-related hearing loss, remains often uncertain. To assess such data worldwide, we reviewed the literature and selected relevant papers according to the following main criteria: the methodology of the *SLC26A4* gene analysis, implying sequencing of all coding exons of *SLC26A4* with flanking regions, which allowed us to conclude the presence or absence of c.919-2A>G in the studied patients (at that, more than two unrelated families were studied) and a mandatory indication of the territorial affiliation and/or the ethnicity of patients. In addition, if the required information was not available, we calculated ourselves the proportion of alleles carrying c.919-2A>G among all mutant *SLC26A4* alleles identified in patients. Based on the obtained data, we came to the conclusion that the spatial distribution of c.919-2A>G can be limited to the territory of Eurasia, since c.919-2A>G was not found on other world continents, as follows from the relevant studies [38,39,40,41,42,43,44,45,46]. Figure 2 represents a hot map demonstrating the c.919-2A>G prevalence in patients with *SLC26A4*-related hearing loss in Eurasia.

The highest frequency of c.919-2A>G in patients with hearing loss is observed in China and Mongolia. The *SLC26A4* pathogenic variants are the second-most common cause of deafness in China. The data on the c.919-2A>G prevalence were obtained for patients of Han Chinese ethnicity from various regions of China as well as for some minor ethnic groups (Hui, Uighur, Tibetan, Tu, Mongolian, etc.) living in China. Numerous studies revealed that the frequency of c.919-2A>G detected in patients sufficiently exceeds 40%, reaching 60–70% in some regions of China [4,12,47,48,49,50,51,52,53,54,55]. The c.919-2A>G variant is observed with frequency in the range of 60–70% in Mongolian patients from Mongolia and Mongolians living in the northwest of China [22,25]. In Korea and Japan, the c.919-2A>G appears to be the second-most common, by frequency, pathogenic *SLC26A4* variant (after c.2168A>G (p.His723Arg)) in patients with hearing loss, and its frequency falls within 20–40% in Korea and 5–10% in Japan, respectively [5,11,19,56,57]. In Thailand, c.919-2A>G was found in one third of all mutated *SLC26A4* alleles in a small sample of patients with Pendred syndrome [58]. The c.919-2A>G has also been found in several Iranian families [9], as well as in Turkish families; thus, the “area” of c.919-2A>G apparently extends to Turkey as a result of historical migration of Turks from Central Asia to Anatolia [59].

The detection of c.919-2A>G in multiple patients from different Asian populations suggests that it might have arisen on a common ancestral founder chromosome. To our knowledge, there are only a few studies aimed at confirming this hypothesis by analyzing the genetic background (haplotypes) of c.919-2A>G [11,12,18,60]. The study by Park et al. [11] was the first study where haplotypes bearing c.919-2A>G were analyzed: three STRs (D7S496, D7S2459, and D7S2456) were used for haplotype analysis in several probands of different ethnicities (Korean, Chinese, and Japanese) who were homozygous or heterozygous for c.919-2A>G. The authors did not reveal a strong association of certain STR alleles with c.919-2A>G on different chromosomes and suggested that c.919-2A>G may be an older founder mutation that has undergone ancestral recombination events with the flanking STR markers. Nevertheless, they did not rule out that c.919-2A>G is a hot spot for recurrent mutational events, despite this allele not being observed in western populations [11]. Subsequently, Yang et al. (2005) analyzed the c.919-2A>G associated haplotypes by the genotyping of five STRs (D7S2549, D7S2420, D7S496, D7S2459, and D7S2456) in four Taiwanese families [18]. Haplotype analysis showed a significant haplotype between markers D7S2420 and D7S2456 common to the family members carrying c.919-2A>G, suggesting that they may be derived from a common ancestor [18]. In the study by Reiisi et al. (2014), different STR haplotypes (defined by the specific alleles of D7S2420, D7S496, D7S2459, and D7S2456) were revealed in two Iranian families carrying *SLC26A4* variants c.919-2A>G or c.416G>T (p.Gly139Val) in each of them: *2-2-1-2* for the c.919-2A>G-associated haplotype in one family and *1-1-1-1* for the c.416G>T (p.Gly139Val)-associated haplotype in another family [60]. In the study by Wu et al. (2005), the evidence of a common ancestral origin for c.919-2A>G was also obtained, since on the majority of chromosomes with c.919-2A>G in patients homozygous or heterozygous for c.919-2A>G from Taiwan (Han Chinese), the core haplotype consisting of four SNPs closely flanking c.919-2A>G (JST160568, JST089508, JST160566, and JST160565) was revealed [12].

In our recent study [27], we revealed a high rate of the *SLC26A4*-related hearing loss in Tuvinian patients belonging to indigenous Siberian people living in Southern Siberia (Russia). At that, we found that the frequency of c.919-2A>G reaches 69.3% among all *SLC26A4* mutant alleles identified in Tuvinian patients, which allowed us to suggest a role of the founder effect in the accumulation of c.919-2A>G in these indigenous Siberian people.

To evaluate a presumable common origin of c.919-2A>G in Tuvinians, we performed haplotype analysis by the genotyping of polymorphic genetic markers (STRs and SNPs) both within and flanking the *SLC26A4* gene in homozygous carriers of this *SLC26A4* pathogenic variant. Our choice of analyzed five STRs (D7S2420, D7S496, D7S2459, D7S2456, and D7S525), surrounding c.919-2A>G, was based on their use in previous studies in the haplotype analysis for several recurrent pathogenic *SLC26A4* variants: c.707T>C (p.Leu236Pro) and c.1246A>C (p.Thr416Pro) in families originating from Western Europe and the USA [61]; c.2168A>G (p.His723Arg) in Korean and Japanese families, c.2027T>A (p.Leu676Gln) in Mongolian patients, and c.269C>T (p.Ser90Leu) in Pakistani patients [11]; c.412G>T (p.Val138Phe) and c.85G>C (p.Glu29Gln) in German and Danish patients [13,62]; c.919-2A>G in patients of Asian origin [11,18] and in Iranian families [60]; c.1541A>G (p.Gln514Arg) in Spanish patients [14]; c.416G>T (p.Gly139Val) in Iranian families [60]; c.716T>A (p.Val239Asp) in Pakistani and Iranian patients [16,63]; and c.1003T>C (p.Phe335Leu), c.1554G>A (p.Trp518Ter), c.84C>A (p.Ser28Arg), and c.2235+2T>C in Brazilian patients [38]. In addition, in the study by Mojtabavi Naeini et al. [64], the characteristics and the allelic and haplotype frequencies of D7S2420, D7S496, and D7S2459 were examined in five ethnic groups (Fars, Azari, Turkmen, Gilaki, and Arab) of the Iranian population. We revealed the *278-120-147-244-227* haplotype (D7S2420-D7S496-/c.919-2A>G/-D7S2459-D7S2456-D7S525), encompassing about 2.8 Mb, in the majority of mutant chromosomes bearing c.919-2A>G (91.3%) (Table 1). This haplotype, as well as the other three STR haplotypes found in homozygotes for c.919-2A>G, was absent in the control sample, which emphasizes the specificity of the genetic background for c.919-2A>G in Tuvinians.

In addition, we genotyped nine intragenic SNPs flanking c.919-2A>G and found the only haplotype A-C-T-A-G-G-C-A-C constituted by the specific allelic combination of all SNPs (rs2248464-rs2248465-rs3801943-rs2712212-/c.919-2A>G/-rs2395911-rs2712211-rs3801940-rs2072064-rs2072065), encompassing 31.039 kb, in all homozygotes for c.919-2A>G, while the frequency of this haplotype reached only 2.8% in the control sample (Table 1).

Thus, based on the common STR and SNP haplotypes bearing c.919-2A>G found in Tuvinians, we obtained convincing evidence supporting the origin of c.919-2A>G from a single ancestor, and the observed accumulation of c.919-2A>G in this indigenous Siberian people may be explained by the founder effect.

In addition, we roughly estimated the potential time intervals of the c.919-2A>G occurrence in the Tuva. As far as we know, there are no age estimations for any pathogenic variants of the *SLC26A4* gene yet, and the age of c.919-2A>G was evaluated by us for the first time. It is worth noting that the methods applied for the estimation of the age of mutation are sensitive to demographic parameters (Appendix A) [36,37,65,66,67]. In view of the lack of reliable data on the variation of the population size of Tuvinians throughout their history, the time of the c.919-2A>G occurrence in the Tuva territory should be considered only as an approximate value. Nevertheless, the partly overlapping time intervals obtained at different population growth rates (d = 0.05, d = 0.1, and d = 0.5) are almost coincided for STR markers (2575–4950 years, 1575–2675 years, and 875–1475 years) and SNP markers (2275–4775 years, 1325–2575 years, and 725–1350 years) (Table 2).

Now, Tuvinians live mainly in the Tyva Republic (Tuva) located in Southern Siberia (Russia), which is bordered by Mongolia in the south and the east. Besides the Tyva Republic, relatively small groups of Tuvinians also live in the northern part of Mongolia and in the Xinjiang Uygur Autonomous Region of China [68,69]. Tuva is located in the geographical center of the Asian continent, and the ancient population of Tuva experienced different gene flows from neighboring regions. At different times, Tuva was at the periphery of a powerful state of Huns (the 2nd century BC—the 1st century AD) or was incorporated in the Ancient Turkic Khaganate (the 6th–8th centuries), the Uyghur Khaganate (the 8th–9th centuries), the Yenisei Kyrgyz Khaganate (the 9th–12th centuries), and also in the Mongol Empire (the 13th–14th centuries). These historical events had a certain impact on the formation of the Tuvinian ethnic group [70,71]. We believe that c.919-2A>G could have appeared in the ancestors of the modern Tuvinian population as a result of different gene flows before the final formation of the Tuvinian ethnos, which was completed by the end of the 13th–14th centuries [70,71].

A very interesting finding of our study was the identity of the “internal” haplotype A-G-G-C (rs2712212-/c.919-2A>G/-rs2395911-rs2712211-rs3801940), encompassing ~4.5 kb, found in the c.919-2A>G homozygotes from Tuva (Tuvinians) and the core haplotype (formed by the same SNPs) in the c.919-2A>G carriers from Taiwan (Han Chinese) [12]. This finding indicates the common ancestor for “Tuvinian” and “Chinese” founder chromosomes with c.919-2A>G. Thus, we speculate that c.919-2A>G could have arisen in the geographically close territories of China or Tuva and subsequently spread to other regions of Asia.

## 5. Conclusions

The common STR and SNP haplotypes carrying c.919-2A>G, found in Tuvinian patients, convincingly indicate the origin of this *SLC26A4* pathogenic variant from a common ancestor that supports a crucial role of the founder effect in the accumulation of c.919-2A>G in the indigenous Siberian people living in Southern Siberia. The identity of small haplotype (~4.5 kb) bearing c.919-2A>G found in Tuvinian and Han Chinese carriers of c.919-2A>G indicates their common founder chromosomes with c.919-2A>G. The *SLC26A4* pathogenic variant c.919-2A>G could have arisen in the geographically close territories of China or Tuva and subsequently spread to other regions of Asia.

## Figures and Tables

**Figure 1 genes-14-00928-f001:**
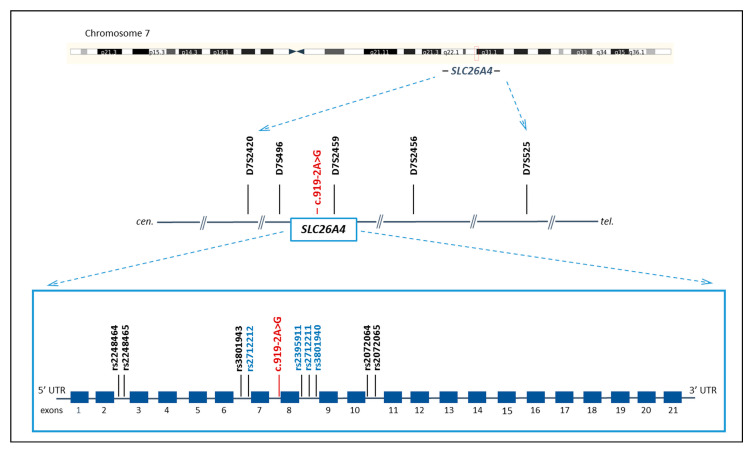
Schematic structure of the *SLC26A4* gene and the location of genetic markers (five STRs and nine SNPs) that were used to reconstruct the c.919-2A>G haplotypes. Location of *SLC26A4* gene is shown by red square. The c.919-2A>G variant is marked by red color. Four of SNP markers from the study by Wu et al. [12] are marked by blue color.

**Figure 2 genes-14-00928-f002:**
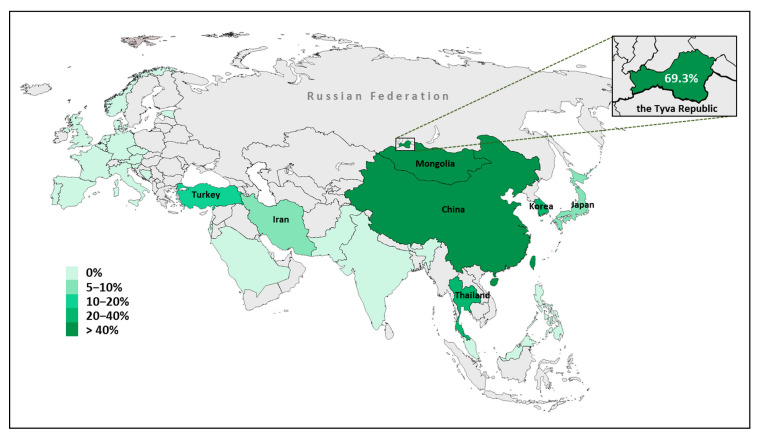
The hot map demonstrating the proportion of c.919-2A>G among all mutated *SLC26A4* alleles revealed in patients with *SLC26A4*-related hearing loss in the territory of Eurasia. The geographic regions for which no data are available are marked by gray color.

**Table 1 genes-14-00928-t001:** The frequencies of STR and SNP haplotypes found among the chromosomes bearing c.919-2A>G, in comparison with the normal chromosomes.

STR Haplotypes D7S2420-D7S496-/c.919-2A>G/-D7S2459-D7S2456-D7S525 (~2.8 Mb)	Frequency of Haplotypes		X^2^	*p*
	Mutant Chromosomes	Normal Chromosomes		
** *278-120-147-244-227* **	0.9130	0.0	150	**<10^−35^**
*278-120-147-244-229*	0.0435	0.0	2.4	0.0704
*278-120-147-244-221*	0.0217	0.0	0.28	0.2674
*278-120-147-244-225*	0.0217	0.0	0.28	0.2674
Other haplotypes	0.0	1.0	-	-
**SNP Haplotypes** **rs2248464-rs2248465-rs3801943-rs2712212*-/c.919-2A>G/-rs2395911*-rs2712211*-rs3801940*-rs2072064-rs2072065** **(31.039 kb)**	**Frequency of Haplotypes**		**X^2^**	** *p* **
	**Mutant** **Chromosomes**	**Normal** **Chromosomes**		
**A-C-T-A-G-G-C-A-C**	1.0	0.0280	150	**<10^−36^**
Other haplotypes	0.0	0.9720	-	-

Designations of the STR alleles included in haplotypes correspond to the size of the PCR products (in nucleotides). The most common haplotypes are shown in bold. *—rs2712212, rs2395911, rs2712211, and rs3801940 correspond to SNPs analyzed in the study by Wu et al. [12]. The haplotype A-G-G-C (rs2712212*-rs2395911*-rs2712211*-rs3801940*) is underlined. Its allelic composition corresponds to the core haplotype T-C-C-G in the study by Wu et al. [12]. Statistically significant (*p* < 0.05) differences in haplotype frequencies are in bold.

**Table 2 genes-14-00928-t002:** The results of the c.919-2A>G age dating.

Genetic Markers Used for Calculations	d	The Single-Marker Method		The DMLE + Calculation	
		g	Age	g (95% CI)	Age (95% CI)
STR markers *	0.05	22	550 years	103–198	2575–4950 years
	0.1	21	525 years	63–107	1575–2675 years
	0.2	17	425 years	35–59	875–1475 years
SNP markers	0.05	-	-	91–191	2275–4775 years
	0.1			53–103	1325–2575 years
	0.2			29–54	725–1350 years

*—The distal STR marker D7S525 was used for the c.919-2A>G age estimation by the single-marker method, and the STR haplotypes were used for c.919-2A>G age estimation by the DMLE+ v.2.3 program. d—population growth rate. g—the number of generations; the age of mutation was calculated as g × 25 years. CI—confidence interval.

## Data Availability

The data presented in this study are available in this article and Appendix A.

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
