# Peer review of "Insight into the Natural History of Pathogenic Variant c.919-2A>G in the SLC26A4 Gene Involved in Hearing Loss: The Evidence for Its Common Origin in Southern Siberia (Russia)"

_genes, 2023, doi:10.3390/genes14040928_

Round 1

Reviewer 1 Report

In this study, the authors investigated the haplotype of SLC26A4 c.919-2A>G mutation in Tuva by analyzing STR markers and SNPs and compared it with normal controls. The results showed that the haplotype comprising four out of five STR markers or all nine SNPs was identical in all 23 patients (46 chromosomes). A small haplotype constituted by four SNPs was also consistent with the c.919-2A>G haplotype in Taiwan. Moreover, the authors estimated the age of onset of c.919-2A>G from these haplotypes. Based on these findings, the authors concluded that c.919-2A>G originated from a common ancestor in China Tuva and spread to other regions in Asia. In general, this is a well-written manuscript that presents interesting data. I believe that the manuscript needs some modifications.

Minor comments

1.       I suggest that the first part of the title be removed as it may appear superb and does not accurately reflect the contents of the study. The author might think about rephrasing the title. Suggestion: "Evidence for a Common Origin of SLC26A4 c.919-2A>G in Southern Siberia (Russia)"

2.       Figure 2 should be revised for the following reasons.

-       The axes' titles are missing, which makes it difficult for readers to understand what the graph depicts.

-       The font size of the axis labels is too small, which may make it challenging for readers to read the information presented.

-       The graph is presented in a 3D bar chart format, making it harder for readers to read exact values.

I recommend that the authors consider these issues and make the necessary revisions to improve the clarity and readability of the figure.

3.       In figure 3, it is unclear what "c.919-2A>G frequency" means. Is this the frequency not in patients with hearing loss but in patients with SLC26A4-related hearing loss? Please clarify it.

Author Response

Reviewer 1

Comments and Suggestions for Authors

In this study, the authors investigated the haplotype of SLC26A4 c.919-2A>G mutation in Tuva by analyzing STR markers and SNPs and compared it with normal controls. The results showed that the haplotype comprising four out of five STR markers or all nine SNPs was identical in all 23 patients (46 chromosomes). A small haplotype constituted by four SNPs was also consistent with the c.919-2A>G haplotype in Taiwan. Moreover, the authors estimated the age of onset of c.919-2A>G from these haplotypes. Based on these findings, the authors concluded that c.919-2A>G originated from a common ancestor in China Tuva and spread to other regions in Asia. In general, this is a well-written manuscript that presents interesting data. I believe that the manuscript needs some modifications.

Minor comments

  1. I suggest that the first part of the title be removed as it may appear superb and does not accurately reflect the contents of the study. The author might think about rephrasing the title. Suggestion: "Evidence for a Common Origin of SLC26A4 c.919-2A>G in Southern Siberia (Russia)"

Authors: Thank you for this suggestion. Nevertheless, we believe that the original title "Insight into the natural history of pathogenic SLC26A4 variants involved in hearing loss: the evidence for a common origin of c.919-2A>G in Southern Siberia (Russia)" is more appropriate, because, along with our data on the common origin of c.919-2A>G in a local region (the Tyva Republic, Southern Siberia), we discuss the prevalence of c.919-2A>G, as well as (albeit to a lesser extent) of other pathogenic SLC26A4 variants in world populations.

  1. Figure 2 should be revised for the following reasons.

- The axes' titles are missing, which makes it difficult for readers to understand what the graph depicts.

- The font size of the axis labels is too small, which may make it challenging for readers to read the information presented.

- The graph is presented in a 3D bar chart format, making it harder for readers to read exact values.

I recommend that the authors consider these issues and make the necessary revisions to improve the clarity and readability of the figure.

Authors: Your critical comment drew our attention to Figure 2 and we decided to remove this Figure due to its redundancy as it shows data already presented in Supplementary Table 3.

  1. In figure 3, it is unclear what "c.919-2A>G frequency" means. Is this the frequency not in patients with hearing loss but in patients with SLC26A4-related hearing loss? Please clarify it.

Authors: We clarified this issue by correction of the caption of this Figure to “The hot map demonstrating the proportion of c.919-2A>G among all mutated SLC26A4 alleles revealed in patients with SLC26A4-related hearing loss on the territory of Eurasia.” In addition, we slightly corrected the corresponding part of text (lines 260-267 in the Clean version of manuscript).

Reviewer 2 Report

The manuscript entitled “Insight into the Natural History of Pathogenic SLC26A4 Variants Involved in Hearing Loss: the Evidence for a Common Origin of c.919-2A>G in Southern Siberia (Russia)” by Danilchenko V.Y. et al., presents results of haplotype analyses in patients homozygous for c.919-2A>G and in healthy controls. . Data provided in the reviewed manuscript may be genetically relevant, but the following questions and concerns should be addressed before publication:

Major comments:

The manuscript requires the change in the organization of the text. Data should not been duplicated in different parts of the manuscript – e.g. information on frequency of the c.919-2A>G variant. Some information provided in the Results section should be moved to the Materials and Methods section (e.g. Selection and genotyping of STR and SNP markers) or to the Discussion section (e.g. discussion of the results in the context of literature data (Wu et al.))

The designation of haplotypes identified in STR analyses should be clarified. What does 278, 120, 147, 144, 227 etc. means? It should be clearly stated that those are the size of the PCR products. Please consider converting the size of the products to the number of repeats. Names of the identified haplotypes should be unified with those described in the literature and discussed in the reviewed manuscript.

The phrase “relatively “old” mutation in unclear – please rephrase this sentences through the manuscript. It would be beneficial to discuss the theoretical age of c.919-2A>G variant in the context of other known founder alleles.

Minor comments:

The Authors should add some information on characteristic SLC26A4-related inner ear malformations (EVA/IP2) in the introduction section.

Figure 2 is hard to analyze/interpret. Please change it to standard grouped barplot. Descriptions of the axes are needed.

The legend or caption of the figure 3 should contain explanation of the gray color.

Tables footnotes should include an explanation of all abbreviations contained in tables.

Please consider the deletion of the words “Southern Siberia” and “Russia” from keywords section.

Different names of the SLC26A4 pathogenic variants should be unified through the text. Please describe them on cDNA and protein levels (in relation to one NM and NP reference sequences).

Explain all abbreviations in appropriate sections of the manuscript – when an abbreviation appear for the first time e.g. STR and SNP in line 90.

Please provide exact numbers to presented percentages, e.g. 71% (99/139)

There are some grammar errors and typos throughout the manuscript. The manuscript would benefit from proof reading. 

Author Response

Reviewer 2

Comments and Suggestions for Authors

The manuscript entitled “Insight into the Natural History of Pathogenic SLC26A4 Variants Involved in Hearing Loss: the Evidence for a Common Origin of c.919-2A>G in Southern Siberia (Russia)” by Danilchenko V.Y. et al., presents results of haplotype analyses in patients homozygous for c.919-2A>G and in healthy controls. . Data provided in the reviewed manuscript may be genetically relevant, but the following questions and concerns should be addressed before publication:

Major comments:

The manuscript requires the change in the organization of the text. Data should not been duplicated in different parts of the manuscript – e.g. information on frequency of the c.919-2A>G variant. Some information provided in the Results section should be moved to the Materials and Methods section (e.g. Selection and genotyping of STR and SNP markers) or to the Discussion section (e.g. discussion of the results in the context of literature data (Wu et al.))

Authors: We are grateful to you for this very helpful comment. We modified the structure of our manuscript according to your suggestion and now it seems more logical and consistent.

The designation of haplotypes identified in STR analyses should be clarified. What does 278, 120, 147, 144, 227 etc. means? It should be clearly stated that those are the size of the PCR products. Please consider converting the size of the products to the number of repeats. Names of the identified haplotypes should be unified with those described in the literature and discussed in the reviewed manuscript.

Authors: In our study, as in many other studies, the designation of STR haplotypes is based on the composition of certain alleles forming them, in turn, the name of a certain allele corresponds to the size (in nucleotides) of the PCR product, for example, 278, 120, 147, 144, 227, etc. This is the standard approach for classifying STR alleles if their detection was carried out by the fragment analysis. In studies where STR markers are genotyped by electrophoresis, the names of the STR alleles are usually assigned numerically, for example, allele 1, allele 2, and so on. It should also be noted that the analysis of STR markers by fragment analysis does not allow estimating the number of repeats.

The phrase “relatively “old” mutation in unclear – please rephrase this sentences through the manuscript. It would be beneficial to discuss the theoretical age of c.919-2A>G variant in the context of other known founder alleles.

Authors: We modified the relevant part of the Discussion section and discussed the age of c.919-2A>G in relation to ethnic history of Tuvinians (lines 357-389 in the Clean version of manuscript). As far as we know, there are no age estimations for any pathogenic variants of the SLC26A4 gene yet. The age of c.919-2A>G was evaluated by us for the first time.

Minor comments:

The Authors should add some information on characteristic SLC26A4-related inner ear malformations (EVA/IP2) in the introduction section.

Authors: We added necessary information in the Introduction section (lines 43-46 in the Clean version of manuscript).

Figure 2 is hard to analyze/interpret. Please change it to standard grouped barplot. Descriptions of the axes are needed.

Authors: Your critical comment drew our attention to Figure 2 and we decided to remove this Figure due to its redundancy as it shows data already presented in Supplementary Table 3.

The legend or caption of the figure 3 should contain explanation of the gray color.

Authors: We added to the legend of Figure 2 (old Figure 3): “The geographic regions for which no available data are marked by gray color.”

Tables footnotes should include an explanation of all abbreviations contained in tables.

Authors: We added missing information for all abbreviations to Tables’ footnotes.

Please consider the deletion of the words “Southern Siberia” and “Russia” from keywords section.

Authors: Fixed. We removed “Southern Siberia” and “Russia” from keywords.

Different names of the SLC26A4 pathogenic variants should be unified through the text. Please describe them on cDNA and protein levels (in relation to one NM and NP reference sequences).

Authors: Fixed. We carefully checked and unified the designations of all variants throughout the text.

Explain all abbreviations in appropriate sections of the manuscript – when an abbreviation appear for the first time e.g. STR and SNP in line 90.

Authors: Fixed. We added the explanations for these abbreviations where they appear for the first time (lines 59-60 in the Clean version of manuscript): STRs (short tandem repeats) and SNPs (single nucleotide polymorphisms).

Please provide exact numbers to presented percentages, e.g. 71% (99/139)

Authors: Fixed. We have added exact numbers to presented percentages where appropriate.

There are some grammar errors and typos throughout the manuscript. The manuscript would benefit from proof reading.

Authors: We have thoroughly proofread the manuscript with the help of a native English speaking colleague and tried to correct all grammatical errors and typos.

Reviewer 3 Report

Presented paper represents a very serios fundamental work. Authors aimed to test a presumable common origin of c.919-2A>G in Tuvinians by analyzing the genetic background (haplotypes) of c.919-2A>G in the carriers of this SLC26A4 pathogenic variant. It is proposed that c.919-2A>G, apparently, is a relatively “old” mutation that could have arisen in geographically close territories of China or Tuva and subsequently spread to other regions of Asia.

Comments

Paper is too big and hard to read due to many explanations in brackets.

In the introduction: lines 45-52 does not consist references.

There is also not mentioned the paper of Russian colleagues

Mironovich OL, Bliznetz EA, Markova TG, Geptner EN, Lalayants MR, Zelikovich EI, Tavartkiladze GA, Polyakov AV. [Results of molecular genetic testing in Russian patients with Pendred syndrome and allelic disorders]. Genetika. 2017 Jan;53(1):88-99. Russian. PMID: 29372807.

Author Response

Reviewer 3

Comments and Suggestions for Authors

Presented paper represents a very serios fundamental work. Authors aimed to test a presumable common origin of c.919-2A>G in Tuvinians by analyzing the genetic background (haplotypes) of c.919-2A>G in the carriers of this SLC26A4 pathogenic variant. It is proposed that c.919-2A>G, apparently, is a relatively “old” mutation that could have arisen in geographically close territories of China or Tuva and subsequently spread to other regions of Asia.

Comments

Paper is too big and hard to read due to many explanations in brackets.

Authors: We modified some parts of the text trying to make our paper more logical and consistent.

In the introduction: lines 45-52 does not consist references.

Authors: Fixed. We have added necessary references (line 54 in the Clean version of manuscript).

There is also not mentioned the paper of Russian colleagues

Mironovich OL, Bliznetz EA, Markova TG, Geptner EN, Lalayants MR, Zelikovich EI, Tavartkiladze GA, Polyakov AV. [Results of molecular genetic testing in Russian patients with Pendred syndrome and allelic disorders]. Genetika. 2017 Jan;53(1):88-99. Russian. PMID: 29372807.

Authors: We know about this study. Unfortunately, there is no information in this study that meets the selection criteria for data on the proportion of c.919-2A>G in patients (lines 260-265 in the Clean version of manuscript).

After minor revision paper could be published in the journal

Round 2

Reviewer 2 Report

Thank you for all the corrections. The article has been sufficiently improved. 
Some minor points should be addresed before publication:

1. The title of the manuscript "Insight into the Natural History of Pathogenic SLC26A4 Variants Involved in Hearing Loss: the Evidence for a Common Origin of c.919-2A>G in Southern Siberia (Russia)" should be edited. The natural history of only one variant is thorougly disscussed in the Manuscript. 

2. Please add to the table 2 explanation of 278, 120, 147, 144, 227 numbers. It should be clearly stated that those are the size of the PCR products.

Author Response

Reviewer 2

Thank you for all the corrections. The article has been sufficiently improved. 
Some minor points should be addresed before publication:

  1. The title of the manuscript "Insight into the Natural History of Pathogenic SLC26A4 Variants Involved in Hearing Loss: the Evidence for a Common Origin of c.919-2A>G in Southern Siberia (Russia)" should be edited. The natural history of only one variant is thorougly disscussed in the Manuscript. 

Authors: We modified the title of manuscript according to your suggestion: “Insight into the Natural History of Pathogenic Variant c.919-2A>G in the SLC26A4 Gene Involved in Hearing Loss: the Evidence for its Common Origin in Southern Siberia (Russia)”.

  1. Please add to the table 2 explanation of 278, 120, 147, 144, 227 numbers. It should be clearly stated that those are the size of the PCR products

Authors: We added to the footnote of Table 1: “Designations of the STR alleles included in haplotypes correspond to the size of the PCR products (in nucleotides).”